# RexUIE: A Recursive Method with Explicit Schema Instructor for Universal Information Extraction

**Chengyuan Liu**[1*]
liucy1@zju.edu.cn

**Fubang Zhao**[2*]
fubang.zfb@alibaba-inc.com

**Yangyang Kang**[2†]
yangyang.kangyy@alibaba-inc.com

**Jingyuan Zhang**[2]
zhangjingyuan1994@gmail.com

**Xiang Zhou**[3]
0020355@zju.edu.cn

**Changlong Sun**[2]
changlong.scl@taobao.com

**Kun Kuang**[1†]
kunkuang@zju.edu.cn

**Fei Wu**[1,4]
wufei@zju.edu.cn

[1]College of Computer Science and Technology, Zhejiang University, [2]Damo Academy, Alibaba Group
[3]Guanghua Law School, Zhejiang University, [4]Shanghai Institute for Advanced Study of Zhejiang University

## Abstract

Universal Information Extraction (UIE) is an area of interest due to the challenges posed by varying targets, heterogeneous structures, and demand-specific schemas. Previous works have achieved success by unifying a few tasks, such as Named Entity Recognition (NER) and Relation Extraction (RE), while they fall short of being true UIE models particularly when extracting other general schemas such as quadruples and quintuples. Additionally, these models used an implicit structural schema instructor, which could lead to incorrect links between types, hindering the model's generalization and performance in low-resource scenarios. In this paper, we redefine the true UIE with a formal formulation that covers almost all extraction schemas. To the best of our knowledge, we are the first to introduce UIE for any kind of schemas. In addition, we propose RexUIE, which is a **R**ecursive Method with **Ex**plicit Schema Instructor for **UIE**. To avoid interference between different types, we reset the position ids and attention mask matrices. RexUIE shows strong performance under both full-shot and few-shot settings and achieves state-of-the-art results on the tasks of extracting complex schemas.

## 1 Introduction

As a fundamental task of natural language understanding, Information Extraction (IE) has been widely studied, such as Named Entity Recognition (NER), Relation Extraction (RE), Event Extraction (EE), Aspect-Based Sentiment Analysis (ABSA), etc. However, the task-specific model structures hinder the sharing of knowledge and structure within the IE community.

Some recent studies attempt to model NER, RE, and EE together to take advantage of the dependencies between subtasks. Lin et al. (2020); Nguyen

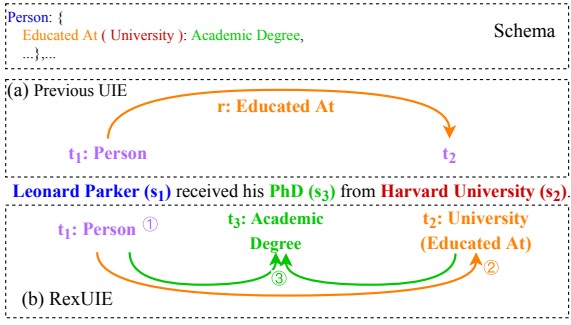

Figure 1: Comparison of RexUIE with previous UIE. (a) The previous UIE models the information extraction task by defining the text spans and the relation between span pairs, but it is limited to extracting only two spans. (b) Our proposed RexUIE recursively extracts text spans for each type based on a given schema, and feeds the extracted information to the following extraction.

et al. (2021) modeled the cross-task dependency by Graph Neural Networks. Another successful attempt is Universal Information Extraction (UIE). Lu et al. (2022) designed novel Structural Schema Instructor (SSI) as inputs and Structured Extraction Language (SEL) as outputs, and proposed a unified text-to-structure generation framework based on T5-Large. While Lou et al. (2023) introduced three unified token linking operations and uniformly extracted substructures in parallel, which achieves new SoTAs on IE tasks.

However, they have only achieved limited success by unifying a few tasks, such as Named Entity Recognition (NER) and Relation Extraction (RE), while ignoring extraction with more than 2 spans, such as quadruples and quintuples, thus fall short of being true UIE models. As illustrated in Figure 1 (a), previous UIE can only extract a pair of spans along with the relation between them, while ignoring other qualifying spans (such as location, time, etc.) that contain information related to the two entities and their relation.

Moreover, previous UIE models are short of explicitly utilizing extraction schema to restrict

---

*Equal contribution.
†Corresponding author.

outcomes. The relation *work for* provides a case wherein the subject and object are the *person* and *organization* entities, respectively. Omitting an explicit schema can lead to spurious results, hindering the model's generalization and performance in resource-limited scenarios.

In this paper, we redefine Universal Information Extraction (UIE) via a comprehensive formal framework that covers almost all extraction schemas. To the best of our knowledge, we are the first to introduce UIE for any kind of schema. Additionally, we introduce RexUIE, which is a **R**ecursive Method with **Ex**plicit Schema Instructor for **UIE**. RexUIE recursively runs queries for all schema types and utilizes three unified token-linking operations to compute the results of each query. We construct an Explicit Schema Instructor (ESI), providing rich label semantic information to RexUIE, and assuring the extraction results meet the constraints of the schema. ESI and the text are concatenated to form the query.

Take Figure 1 (b) as an example, given the extraction schema, RexUIE firstly extracts "Leonard Parker" classified as a "Person", then extracts "Harvard University" classified as "University" coupled with the relation "Educated At" according to the schema. Thirdly, based on the extracted tuples ( *"Leonard Parker", "Person"* ) and ( *"Harvard University", "Educated At (University)"* ), RexUIE derives the span "PhD" classified as an "Academic Degree". RexUIE extracts spans recursively based on the schema, allowing extracting more than two spans such as quadruples and quintuples, rather than exclusively limited to pairs of spans and their relation.

We pre-trained RexUIE on a combination of supervised NER and RE datasets, Machine Reading Comprehension (MRC) datasets, as well as 3 million Joint Entity and Relation Extraction (JERE) instances constructed via Distant Supervision. Extensive experiments demonstrate that RexUIE surpasses the state-of-the-art performance in various tasks and outperforms previous UIE models in few-shot experiments. Additionally, RexUIE exhibits remarkable superiority in extracting quadruples and quintuples.

The contributions of this paper can be summarized as follows:

1. We redefine true Universal Information Extraction (UIE) through a formal framework that covers almost all extraction schemas,

rather than only extracting spans and pairs.

2. We introduce RexUIE, which recursively runs queries for all schema types and utilizes three unified token-linking operations to compute the outcomes of each query. It employs explicit schema instructions to augment label semantic information and enhance the performance in low-resource scenarios.

3. We pre-train RexUIE to enhance low-resource performance. Extensive experiments demonstrate its remarkable effectiveness, as RexUIE surpasses not only previous UIE models and task-specific SoTAs in extracting entities, relations, quadruples and quintuples, but also outperforms large language models (such as ChatGPT) under zero-shot setting.

## 2 Related Work

Task-specific models for IE have been extensively studied, including Named Entity Recognition (Lample et al., 2016; Yan et al., 2021a; Wang et al., 2021), Relation Extraction (Li et al., 2022; Zhong and Chen, 2021; Zheng et al., 2021), Event Extraction (Li et al., 2021), and Aspect-Based Sentiment Analysis (Zhang et al., 2021; Xu et al., 2021).

Some recent works attempted to jointly extract the entities, relations and events (Nguyen et al., 2022; Paolini et al., 2021). OneIE (Lin et al., 2020) firstly extracted the globally optimal IE result as a graph from an input sentence, and incorporated global features to capture the cross-subtask and cross-instance interactions. FourIE (Nguyen et al., 2021) introduced an interaction graph between instances of the four tasks. Wei et al. (2020) proposed using consistent tagging schemes to model the extraction of entities and relationships. Wang et al. (2020) extended the idea to a unified matrix representation. TPLinker formulates joint extraction as a token pair linking problem and introduces a novel handshaking tagging scheme that aligns the boundary tokens of entity pairs under each relation type. Another approach that has been used to address joint information extraction with great success is the text-to-text language generation model. Lu et al. (2021a) generated the linearized sequence of trigger words and argument in a text-to-text manner. Kan et al. (2022) purposed to jointly extract information by adding some general or task-specific prompts in front of the text.

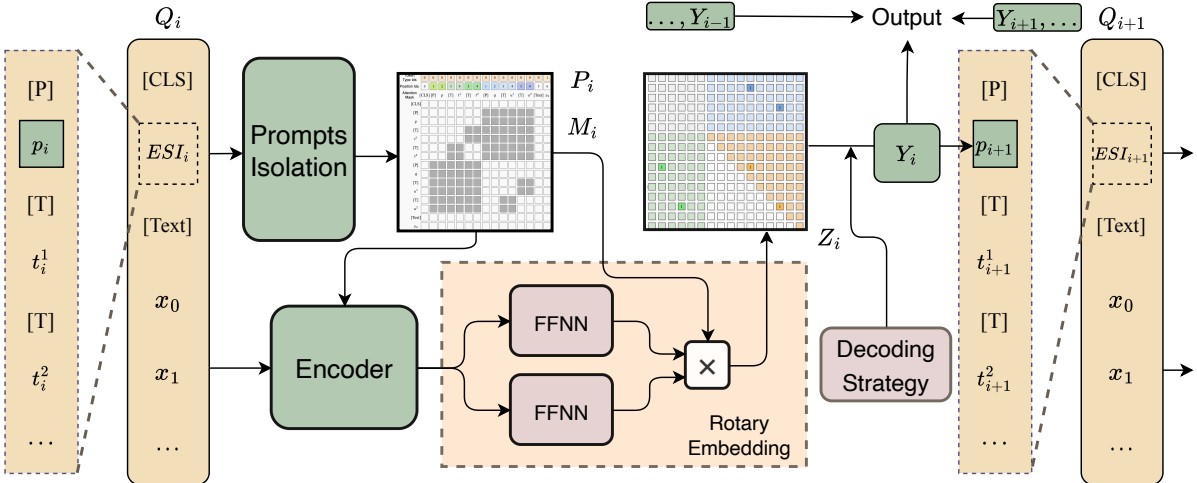

Figure 2: The overall framework of RexUIE. We illustrate the computation process of the $i$-th query and the construction of the $i+1$-th query. $M_i$ denotes the attention mask matrix, and $Z_i$ denotes the score matrix obtained by decoding. $Y_i$ denotes the output of the $i$-th query, with all outputs ultimately combined to form the overall extraction result.

Lu et al. (2022) introduced the unified structure generation for UIE. They proposed a framework based on T5 architecture to generate SEL containing specified types and spans. However, the autoregressive method suffers from low GPU utilization. Lou et al. (2023) proposed an end-to-end framework for UIE, called USM, by designing three unified token linking operations. Empirical evaluation on 4 IE tasks showed that USM has strong generalization ability in zero/few-shot transfer settings.

## 3 Redefine Universal Information Extraction

While Lu et al. (2022) and Lou et al. (2023) proposed Universal Information Extraction as methods of addressing NER, RE, EE, and ABSA with a single unified model, their approaches were limited to only a few tasks and ignored schemas that contain more than two spans, such as quadruples and quintuples. Hence, we redefine UIE to cover the extraction of more general schemas.

In our view, genuine UIE extracts a collection of structured information from the text, with each item consisting of $n$ spans $\mathbf{s} = [s_1, s_2, \ldots, s_n]$ and $n$ corresponding types $\mathbf{t} = [t_1, t_2, \ldots, t_n]$. The spans are extracted from the text, while the types are defined by a given schema. Each pair of $(s_i, t_i)$ is the target to be extracted.

Formally, we propose to maximize the probabil-

ity in Equation 1.

$$
\begin{aligned}
&\prod_{(\mathbf{s},\mathbf{t}) \in \mathbb{A}} p\big((\mathbf{s},\mathbf{t})|\mathbf{C}^n,\mathbf{x}\big) \\
&= \prod_{(\mathbf{s},\mathbf{t}) \in \mathbb{A}} \prod_{i=1}^{n} p\big((s,t)_i \,|\, (\mathbf{s},\mathbf{t})_{<i}, \mathbf{C}^n, \mathbf{x}\big) \\
&= \prod_{i=1}^{n} \left[ \prod_{(s,t)_i \in \mathbb{A}_i|(\mathbf{s},\mathbf{t})_{<i}} p\big((s,t)_i \,|\, (\mathbf{s},\mathbf{t})_{<i}, \mathbf{C}^n, \mathbf{x}\big) \right]
\end{aligned}
\tag{1}
$$

where $\mathbf{C}^n$ denotes the hierarchical schema (a tree structure) with depth $n$, $\mathbb{A}$ is the set of all sequences of annotated information. $\mathbf{t} = [t_1, t_2, \ldots, t_n]$ is one of the type sequences (paths in the schema tree), and $\mathbf{x}$ is the text. $\mathbf{s} = [s_1, s_2, \ldots, s_n]$ denotes the corresponding sequence of spans to $\mathbf{t}$. We use $(s,t)_i$ to denote the pair of $s_i$ and $t_i$. Similarly, $(\mathbf{s},\mathbf{t})_{<i}$ denotes $[s_1, s_2, \ldots, s_{i-1}]$ and $[t_1, t_2, \ldots, t_{i-1}]$. $\mathbb{A}_i|(\mathbf{s},\mathbf{t})_{<i}$ is the set of the $i$-th items of all sequences led by $(\mathbf{s},\mathbf{t})_{<i}$ in $\mathbb{A}$. To more clearly clarify the symbols, we present some examples in Appendix H.

## 4 RexUIE

In this Section, we introduce RexUIE: A Recursive Method with Explicit Schema Instructor for Universal Information Extraction.

RexUIE models the learning objective Equation 1 as a series of recursive queries, with three unified token-linking operations employed to compute the outcomes of each query. The condition $(\mathbf{s},\mathbf{t})_{<i}$

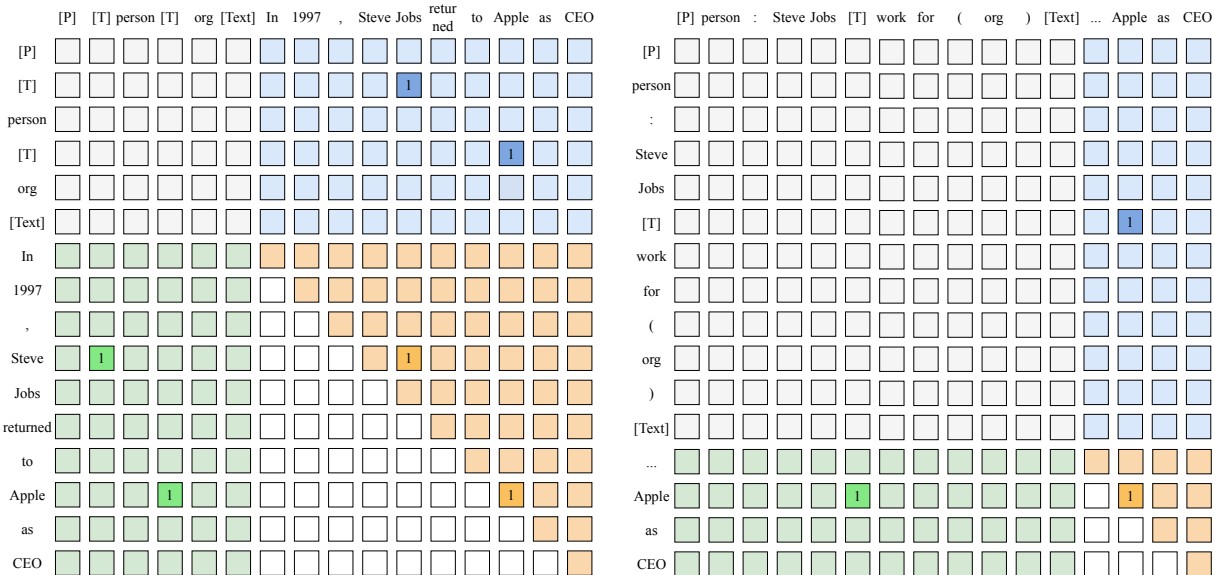

Figure 3: Queries and score matrices for NER and RE. The left sub-figure shows how to extract entities "Steve Jobs" and "Apple". The right sub-figure shows how to extract the relation given the entity "Steve Jobs" coupled with type "person". The schema is organized as *{"person": {"work for (organization)": null}, "organization": null }*. The score matrix is separated into three valid parts: token head-tail, type-token tail and token head-type. The cells scored as 1 are darken, the others are scored as 0.

in Equation 1 is represented by the prefix in the $i$-th query, and $(s,t)_i$ is calculated by the linking operations.

## 4.1 Framework of RexUIE

Figure 2 shows the overall framework. RexUIE recursively runs queries for all schema types. Given the $i$-th query $Q_i$, we adopt a pre-trained language model as the encoder to map the tokens to hidden representations $h_i \in \mathbb{R}^{n \times d}$, where $n$ is the length of the query, and $d$ is the dimension of the hidden states,

$$h_i = \mathbf{Encoder}(Q_i, P_i, M_i) \quad (2)$$

where $P_i$ and $M_i$ denote the position ids and attention mask matrix of $Q_i$ respectively..

Next, the hidden states are fed into two feed-forward neural networks $\mathbf{FFNN}_q, \mathbf{FFNN}_k$ .

Then we apply rotary embeddings following Su et al. (2021, 2022) to calculate the score matrix $Z_i$.

$$Z_i^{j,k} = (\mathbf{FFNN}_q(h_i^j)^\top \mathbf{R}(P_i^k - P_i^j) \\ \mathbf{FFNN}_k(h_i^k)) \otimes M_i^{j,k} \quad (3)$$

where $M_i^{j,k}$ and $Z_i^{j,k}$ denote the mask value and score from token $j$ to $k$ respectively. $P_i^j$ and $P_i^k$ denote the position ids of token $j$ and $k$. $\otimes$ is the

Hadamard product. $\mathbf{R}(P_i^k - P_i^j) \in \mathbb{R}^{d \times d}$ denotes the rotary position embeddings (RoPE), which is a relative position encoding method with promising theoretical properties.

Finally, we decode the score matrix $Z_i$ to obtain the output $Y_i$, and utilize it to create the subsequent query $Q_{i+1}$. All ultimate outputs are merged into the result set $\mathcal{Y} = \{Y_1, Y_2, \dots\}$.

We utilize Circle Loss (Sun et al., 2020; Su et al., 2022) as the loss function of RexUIE, which is very effective in calculating the loss of sparse matrices

$$\mathcal{L}_i = \log(1 + \sum_{\hat{Z}_i^j = 0} e^{\overline{Z}_i^j}) + \log(1 + \sum_{\hat{Z}_i^k = 1} e^{-\overline{Z}_i^k})$$

$$\mathcal{L} = \sum_i \mathcal{L}_i$$

$$(4)$$

where $\overline{Z}_i$ is a flattened version of $Z_i$, and $\hat{Z}_i$ denotes the flattened ground truth, containing only 1 and 0.

## 4.2 Explicit Schema Instructor

The $i$-th query $Q_i$ consists of an Explicit Schema Instructor (ESI) and the text **x**. ESI is a concatenation of a prefix $p_i$ and types $t_i = [t_i^1, t_i^2, \dots]$. The prefix $p_i$ models $(\mathbf{s}, \mathbf{t})_{<i}$ in Equation 1, which is constructed based on the sequence of previously extracted types and the corresponding sequence of spans. $t_i$ specifies what types can be potentially

identified from $\mathbf{x}$ given $p_i$.

We insert a special token [P] before each prefix and a [T] before each type. Additionally, we insert a token [Text] before the text $\mathbf{x}$. Then, the input $Q_i$ can be represented as

$$Q_i = \texttt{[CLS][P]} p_i \texttt{[T]} t_i^1 \texttt{[T]} t_i^2 \ldots \texttt{[Text]} x_0 x_1 \ldots \tag{5}$$

The biggest difference between ESI and implicit schema instructor is that the sub-types that each type can undertake are explicitly specified. Given the parent type, the semantic meaning of each sub-type is richer, thus the RexUIE has a better understanding to the labels.

Some detailed examples of ESI are listed in Appendix I.

### 4.3 Token Linking Operations

Given the calculated score matrix $Z$, we obtain $\tilde{Z}$ from $Z$ by a predefined threshold $\delta$ following

$$\tilde{Z}^{i,j} = \begin{cases} 1 & \text{if } Z^{i,j} \geq \delta \\ 0 & \text{otherwise} \end{cases} \tag{6}$$

Token linking is performed on $\tilde{Z}$, which takes binary values of either 1 or 0 (Wang et al., 2020; Lou et al., 2023). A token linking is established from the $i$-th token to the $j$-th token only if $\tilde{Z}^{i,j} = 1$; otherwise, no link exists. To illustrate this process, consider the example depicted in Figure 3. We expound upon how entities and relations can be extracted based on the score matrix.

**Token Head-Tail Linking** Token head-tail linking serves the purpose of span detection. if $i \leq j$ and $\tilde{Z}^{i,j} = 1$, the span $Q^{i:j}$ should be extracted. The orange section in Figure 3 performs token head-tail linking, wherein both "Steve Jobs" and "Apple" are recognized as entities. Consequently, a connection exists from "Steve" to "Jobs" and another from "Apple" to itself.

**Token Head-Type Linking** Token head-type linking refers to the linking established between the head of a span and its type. To signify the type, we utilize the special token [T], which is positioned just before the type token. As highlighted in the green section of Figure 3, "Steve Jobs" qualifies as a "person" type span, so a link points from "Steve" to the [T] token that precedes "person". Similarly, a link exists from "Apple" to the [T] token preceding "org".

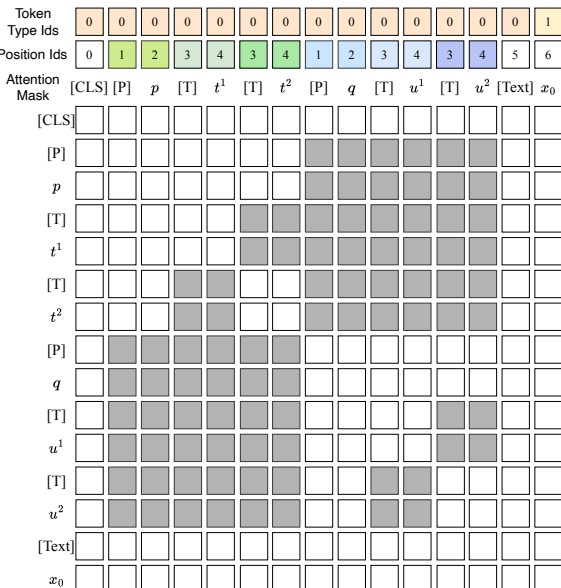

Figure 4: Token type ids, position ids and Attention mask for RexUIE. $p$ and $t$ denote the prefix and types of the first group of previously extracted results. $q$ and $u$ denote the prefix and types for the second group.

**Type-Token Tail Linking** Type-token tail linking refers to the connection established between the type of a span and its tail. Similar to token head-type linking, we utilize the [T] token before the type token to represent the type. As highlighted in the blue section of Figure 3, a link exists from the [T] token preceding "person" to "Jobs" due to the prediction that "Steve Jobs" is a "person" span.

During inference, for a pair of token $\langle i, j \rangle$, if $Z^{i,j} \geq \delta$, and there exists a [T] $k$ that satisfies $Z^{i,k} \geq \delta$ and $Z^{k,j} \geq \delta$, we extract the span $Q^{i:j}$ with the type after $k$.

### 4.4 Prompts Isolation

RexUIE can receives queries with multiple prefixes. To save the cost of time, we put different prefix groups in the same query. For instance, consider the text "Kennedy was fatally shot by Lee Harvey Oswald on November 22, 1963", which contains two "person" entities. We concatenate the two entity spans, along with their corresponding types in the schema respectively to obtain ESI: *[CLS][P]person: Kennedy [T] kill (person) [T] live in (location)...[P] person: Lee Harvey Oswald [T] kill(person) [T] live in (location)....*

However, the hidden representations of type *kill (person)* should not be interfered by type *live in (location)*. Similarly, the hidden representations of prefix *person: Kennedy* should not be interfered by

other prefixes (such as *person: Lee Harvey Oswald*) either.

Inspired by Yang et al. (2022), we present Prompts Isolation, an approach that mitigates interferences among tokens of diverse types and prefixes. By modifying token type ids, position ids, and attention masks, the direct flow of information between these tokens is effectively blocked, enabling clear differentiation among distinct sections in ESI. We illustrate Prompts Isolation in Figure 4. For the attention masks, each prefix token can only interact with the prefix itself, its sub-type tokens, and the text tokens. Each type token can only interact with the type itself, its corresponding prefix tokens, and the text tokens.

Then the position ids $P$ and attention mask $M$ in Equation 3 can be updated. In this way, potentially confusing information flow is blocked. Additionally, the model would not be interfered by the order of prefixes and types either.

### 4.5 Pre-training

To enhance the zero-shot and few-shot performance of RexUIE, we pre-trained RexUIE on the following three distinct datasets:

**Distant Supervision data** $\mathcal{D}_{distant}$  We gathered the corpus and labels from WikiPedia[1], and utilized Distant Supervision to align the texts with their respective labels.

**Supervised NER and RE data** $\mathcal{D}_{superv}$  Compared with $\mathcal{D}_{distant}$, supervised data exhibits higher quality due to its absence of abstract or over-specialized classes, and there is no high false negative rate caused by incomplete knowledge base.

**MRC data** $\mathcal{D}_{mrc}$  The empirical results of previous works (Lou et al., 2023) show that incorporating machine reading comprehension (MRC) data into pre-training enhances the model's capacity to utilize semantic information in prompt. Accordingly we add MRC supervised instances to the pre-training data.

Details of the datasets for pre-training can be found in Appedix G.

## 5 Experiments

We conduct extensive experiments in this Section under both supervised settings and few-shot settings. For implementation, we adopt DeBERTaV3-Large (He et al., 2021) as our text encoder, which

also incorporates relative position information via disentangled attention, similar to our rotary module. We set the maximum token length to 512, and the maximum length of ESI to 256. We split a query into sub-queries when the length of the ESI is beyond the limit. Detailed hyper-parameters are available in Appendix B. Due to space limitation, we have included the implementation details of some experiments in Appendix C.

### 5.1 Dataset

We mainly follow the data setting of Lu et al. (2022); Lou et al. (2023), including ACE04 (Mitchell et al., 2005), ACE05 (Walker et al., 2006), CoNLL03 (Tjong Kim Sang and De Meulder, 2003), CoNLL04 (Roth and Yih, 2004), NYT (Riedel et al., 2013), SciERC (Luan et al., 2018), CASIE (Satyapanich et al., 2020), SemEval-14 (Pontiki et al., 2014), SemEval-15 (Pontiki et al., 2015) and SemEval-16 (Pontiki et al., 2016). We add two more tasks to evaluate the ability of extracting schemas with more than two spans: 1) Quadruple Extraction. We use HyperRED (Chia et al., 2022), which is a dataset for hyper-relational extraction to extract more specific and complete facts from the text. Each quadruple of HyperRED consists of a standard relation triple and an additional qualifier field that covers various attributes such as time, quantity, and location. 2) Comparative Opinion Quintuple Extraction (Liu et al., 2021). COQE aims to extract all the comparative quintuples from review sentences. There are at most 5 attributes for each instance to extract: subject, object, aspect, opinion, and the polarity of the opinion(e.g. better, worse, or equal). We only use the English subset Camera-COQE.

The detailed datasets and evaluation metrics are listed in Appedix A.

### 5.2 Main Results

We first conduct experiments with full-shot training data. Table 1 presents a comprehensive comparison of RexUIE against T5-UIE (Lu et al., 2022), USM (Lou et al., 2023), and previous task-specific models, both in pre-training and non-pre-training scenarios.

We can observe that: 1) RexUIE surpasses the task-specific state-of-the-art models on more than half of the IE tasks even without pre-training. RexUIE exhibits a higher F1 score than both USM and T5-UIE across all the ABSA datasets. Furthermore, RexUIE's performance in the task of

---

[1]https://www.wikipedia.org/

Table 1: F1 result for UIE models with pre-training.

| Dataset | Task-Specific SOTA Methods | | Without Pre-training | | | With Pre-training | | |
|---|---|---|---|---|---|---|---|---|
| | | | T5-UIE | USM | RexUIE | T5-UIE | USM | RexUIE |
| ACE04 | Lou et al. (2022) | **87.90** | 86.52 | 87.79 | **88.02** | 86.89 | 87.62 | 87.25 |
| ACE05-Ent | Lou et al. (2022) | 86.91 | 85.52 | **86.98** | 86.87 | 85.78 | 87.14 | **87.23** |
| CoNLL03 | Wang et al. (2021) | 93.21 | 92.17 | 92.79 | **93.31** | 92.99 | 93.16 | **93.67** |
| ACE05-Rel | Yan et al. (2021b) | **66.80** | 64.68 | 66.54 | 63.44 | 66.06 | **67.88** | 64.87 |
| CoNLL04 | Huguet Cabot and Navigli (2021) | 75.40 | 73.07 | 75.86 | **76.79** | 75.00 | **78.84** | 78.39 |
| NYT | Huguet Cabot and Navigli (2021) | 93.40 | 93.54 | 93.96 | **94.35** | 93.54 | 94.07 | **94.55** |
| SciERC | Yan et al. (2021b) | **38.40** | 33.36 | 37.05 | 38.16 | 36.53 | 37.36 | 38.37 |
| ACE05-Evt-Trg | Wang et al. (2022) | **73.60** | 72.63 | 71.68 | 73.25 | 73.36 | 72.41 | **75.17** |
| ACE05-Evt-Arg | Wang et al. (2022) | 55.10 | 54.67 | 55.37 | **57.27** | 54.79 | 55.83 | **59.15** |
| CASIE-Trg | Lu et al. (2021b) | 68.98 | 68.98 | 70.77 | **72.03** | 68.33 | 71.73 | **73.01** |
| CASIE-Arg | Lu et al. (2021b) | 60.37 | 60.37 | **63.05** | 62.15 | 61.30 | 63.26 | **63.87** |
| 14-res | Zhang et al. (2021) | 72.16 | 73.78 | 76.35 | **76.36** | 74.52 | 77.26 | **77.46** |
| 14-lap | Zhang et al. (2021) | 60.78 | 63.15 | 65.46 | **66.92** | 63.88 | 65.51 | **66.41** |
| 15-res | Xu et al. (2021) | 63.27 | 66.10 | 68.80 | **70.48** | 67.15 | 69.86 | **70.84** |
| 16-res | Xu et al. (2021) | 70.26 | 73.87 | 76.73 | **78.13** | 75.07 | **78.25** | 77.20 |
| HyperRED | Chia et al. (2022) | 66.75 | - | - | **73.25** | - | - | **75.20** |
| Camera-COQE | Liu et al. (2021) | 13.36 | - | - | **32.02** | - | - | **32.80** |

Table 1: F1 result for UIE models with pre-training. ∗-Trg means evaluating models with Event Trigger F1, ∗-Arg means evaluating models with Event Argument F1, while detailed metrics are listed in Appendix B. T5-UIE and USM are the previous SoTA UIE models proposed by Lu et al. (2022) and Lou et al. (2023), respectively.

| | Model | 1-Shot | 5-Shot | 10-Shot | AVE-S |
|---|---|---|---|---|---|
| Entity CoNLL03 | T5-UIE | 57.53 | 75.32 | 79.12 | 70.66 |
| | USM | 71.11 | 83.25 | 84.58 | 79.65 |
| | RexUIE | **86.57** | **89.63** | **90.82** | **89.07** |
| Relation CoNLL04 | T5-UIE | 34.88 | 51.64 | 58.98 | 48.50 |
| | USM | 36.17 | 53.2 | 60.99 | 50.12 |
| | RexUIE | **43.80** | **54.90** | **61.68** | **53.46** |
| Event Trigger ACE05-Evt | T5-UIE | 42.37 | 53.07 | 54.35 | 49.93 |
| | USM | 40.86 | 55.61 | 58.79 | 51.75 |
| | RexUIE | **56.95** | **64.12** | **65.41** | **62.16** |
| Event Argument ACE05-Evt | T5-UIE | 14.56 | 31.20 | 35.19 | 26.98 |
| | USM | 19.01 | 36.69 | 42.48 | 32.73 |
| | RexUIE | **30.43** | **41.04** | **45.14** | **38.87** |
| Sentiment 16-res | T5-UIE | 23.04 | 42.67 | 53.28 | 39.66 |
| | USM | 30.81 | **52.06** | 58.29 | 47.05 |
| | RexUIE | **37.70** | 49.84 | **60.56** | **49.37** |

Table 2: Few-Shot experimental results. AVE-S denotes the average performance over 1-Shot, 5-Shot and 10-Shot.

Event Extraction is remarkably superior to that of the baseline models. 2) Pre-training brings in slight performance improvements. By comparing the outcomes in the last three columns, we can observe that RexUIE with pre-training is ahead of T5-UIE and USM on the majority of datasets. After pre-training, ACE05-Evt showed a significant improvement with an approximately 2% increase in F1 score. This implies that RexUIE effectively utilizes the semantic information in prompt texts and establishes links between text spans and their corresponding types. It is worth noting that the schema of trigger words and arguments in ACE05-Evt is complex, and the model heavily relies on the semantic information of labels. 3) The bottom two rows describe the results of extracting quadruples and quintuples, and they are compared with the SoTA methods. Our model demonstrates significantly superior performance on both HyperRED and Camera-COQE, which shows the effectiveness of extracting complex schemas.

### 5.3 Few-Shot Information Extraction

We conducted few-shot experiments on one dataset for each task, following Lu et al. (2022) and Lou et al. (2023). The results are shown in Table 2.

In general, RexUIE exhibits superior performance compared to T5-UIE and USM in a low-resource setting. Specifically, RexUIE relatively outperforms T5-UIE by 56.62% and USM by 32.93% on average in 1-shot scenarios. The success of RexUIE in low-resource settings can be attributed to its ability to extract information learned during pre-training, and to the efficacy of our proposed query, which facilitates explicit schema learning by RexUIE.

| Task | Model | Precision | Recall | F1 |
|---|---|---|---|---|
| Relation CoNLL04 | GPT-3 | - | - | 18.10 |
| | ChatGPT* | 25.16 | 19.16 | 21.76 |
| | DEEPSTRUCT | - | - | 25.80 |
| | USM | - | - | 25.95 |
| | RexUIE* | 44.95 | 23.22 | **30.62** |
| Entity CoNLLpp | ChatGPT | 62.30 | 55.00 | 58.40 |
| | RexUIE* | 91.57 | 66.09 | **76.77** |

Table 3: Zero-Shot performance on RE and NER. * indicates that the experiment is conducted by ourselves.

| | Model | Precision | Recall | F1 |
|---|---|---|---|---|
| Without Pre-train | T5-UIE | 20.92 | 21.56 | 21.23 |
| | RexUIE | 31.43 | 32.62 | **32.02** |
| With Pre-train | T5-UIE | 24.41 | 25.46 | 24.92 |
| | RexUIE | 34.07 | 31.63 | **32.80** |

Table 4: Extraction results on COQE.

## 5.4 Zero-Shot Information Extraction

We conducted zero-shot experiments on RE and NER comparing RexUIE with other pre-trained models, including ChatGPT[2]. We adopt the pipeline proposed by Wei et al. (2023) for Chat-GPT. We used CoNLL04 and CoNLLpp (Wang et al., 2019) for RE and NER respectively. We report Precision, Recall and F1 in Table 3.

RexUIE achieves the highest zero-shot extraction performance on the two datasets. Furthermore, we analyzed bad cases of ChatGPT. 1) ChatGPT generated words that did not exist in the original text. For example, ChatGPT output a span "Coats Michael", while the original text was "Michael Coats". 2) Errors caused by inappropriate granularity, such as "city in Italy" and "Italy". 3) Illegal extraction against the schema. ChatGPT outputs (*Leningrad, located in, Kirov Ballet*), while "Kirov Ballet" is an organization rather than a location.

## 5.5 Complex Schema Extraction

To illustrate the significance of the ability to extract complex schemas, we designed a forced approach to extract quintuples for T5-UIE, which extracts three tuples to form one quintuple. Details are available in Appendix C.

Table 4 shows the results comparing RexUIE with T5-UIE. In general, RexUIE's approach of directly extracting quintuples exhibits superior performance. Although T5-UIE shows a slight performance improvement after pre-training, it is still

[2]https://openai.com/blog/chatgpt

approximately 8% lower than RexUIE on F1.

## 5.6 Absense of Explicit Schema

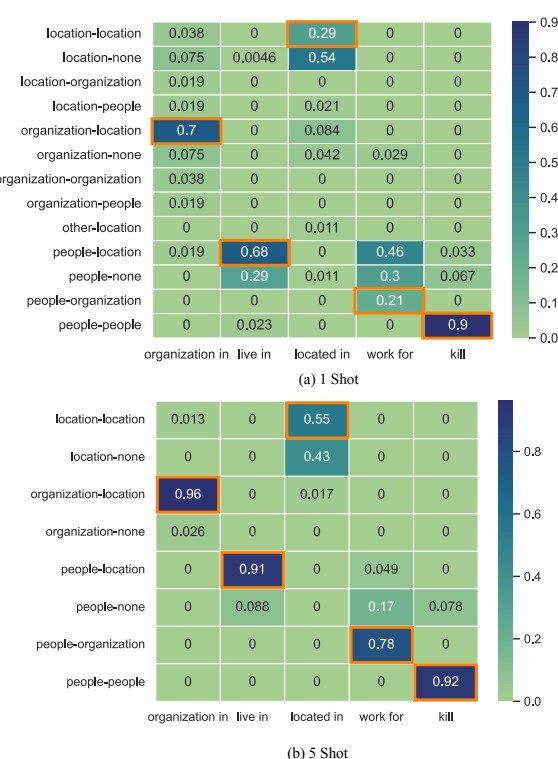

Figure 5: The distribution of relation type versus *subject type-object type* predicted by T5-UIE. We circle the correct cases in orange.

We analyse the distribution of relation type versus *subject type-object type* predicted by T5-UIE as illustrated in Figure 5.

We observe that illegal extractions, such as *person, work for, location*, are not rare in 1-Shot, and a considerable number of subjects or objects are not properly extracted during the NER stage. Although this issue is alleviated in the 5-Shot scenario, we believe that the implicit schema instructor still negatively affects the model's performance.

## 6 Conclusion

In this paper, we introduce RexUIE, a UIE model using multiple prompts to recursively link types and spans based on an extraction schema. We redefine UIE with the ability to extract schemas with any number of spans and types. Through extensive experiments under both full-shot and few-shot settings, we demonstrate that RexUIE outperforms state-of-the-art methods on a wide range of datasets, including quadruples and quintuples extraction. Our empirical evaluation highlights the

significance of explicit schemas and emphasizes that the ability to extract complex schemas cannot be substituted.

## Limitations

Despite demonstrating impressive zero-shot entity recognition and relationship extraction performance, RexUIE currently lacks zero-shot capabilities in events and sentiment extraction due to the limitation of pre-training data. Furthermore, RexUIE is not yet capable of covering all NLU tasks, such as Text Entailment.

## Ethics Statement

Our method is used to unify all of information extraction tasks within one framework. Therefore, ethical considerations of information extraction models generally apply to our method. We obtained and used the datasets in a legal manner. We encourage researchers to evaluate the bias when using RexUIE.

## Acknowledgements

This work was supported in part by National Key Research and Development Program of China (2022YFC3340900), National Natural Science Foundation of China (62376243, 62037001, U20A20387), the StarryNight Science Fund of Zhejiang University Shanghai Institute for Advanced Study (SN-ZJU-SIAS-0010), Alibaba Group through Alibaba Research Intern Program, Project by Shanghai AI Laboratory (P22KS00111), Program of Zhejiang Province Science and Technology (2022C01044), the Fundamental Research Funds for the Central Universities (226-2022-00142, 226-2022-00051).

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

## A Detailed Supervised Datasets for Downstream Tasks

The detailed datasets and evaluation metrics are listed in Table 5. We explain the evaluation metrics as follows.

**Entity Strict F1** An entity mention is correct if its offsets and type match a reference entity.

**Relation Strict F1** A relation is correct if its relation type is correct and the offsets and entity types of the related entity mentions are correct.

**Relation Triplet F1** A relation is correct if its relation type is correct and the string of the related entity mentions are correct.

**Event Trigger F1** An event trigger is correct if its offsets and event type matches a reference trigger.

**Event Argument F1** An event argument is correct if its offsets, role type, and event type match a reference argument mention.

**Sentiment Strict F1** For triples, a sentiment is correct if the offsets of its target, opinion and the sentiment polarity match with the ground truth. For quintuples, a sentiment is correct if the offsets of its subject, object, aspect, opinion and the sentiment polarity match with the ground truth.

**Quadruple Strict F1** A relation quadruple is correct if the relation type and the type and offsets of its subject, object, qualifier match with the ground truth.

| Task | Metric | Dataset |
|---|---|---|
| Entity | Entity Strict F1 | ACE04-Ent |
|  | Entity Strict F1 | ACE05-Ent |
|  | Entity Strict F1 | CoNLL03 |
| Relation | Relation Strict F1 | ACE05-Rel |
|  | Relation Strict F1 | CoNLL04 |
|  | Relation Triplet F1 | NYT |
|  | Relation Strict F1 | SciERC |
| Event | Trigger F1 | ACE05-Evt |
|  | Argument F1 | ACE05-Evt |
|  | Trigger F1 | CASIE |
|  | Argument F1 | CASIE |
| Sentiment | Sentiment Strict F1 | 14-res |
|  | Sentiment Strict F1 | 14-lap |
|  | Sentiment Strict F1 | 15-res |
|  | Sentiment Strict F1 | 16-res |
| Quadruple | Quadruple Strict F1 | HyperRED |
| COQE | Sentiment Strict F1 | Camera-COQE |

Table 5: Detailed supervised datasets and evaluation metrics for each task.

## B Implementation Details

We download the supervised data for pre-training from HuggingFace[3]. For all the downstream datasets, we follow the procedure by Lu et al. (2022); Lou et al. (2023) and then convert them to

---

[3]https://huggingface.co/datasets

|  | Learning Rate | Batch Size | Epoch |
|---|---|---|---|
| Pre-training | 5e-5 | 128 | 5 |
| Low-resource | 1e-5, 3e-5 | 16 | 50, 100 |
| Entity | 1e-5, 3e-5 | 64 | 100, 200 |
| Relation | 1e-5, 3e-5 | 64, 128 | 50, 100, 200 |
| Event | 3e-5 | 64, 96 | 50, 100 |
| Sentiment | 3e-5 | 32 | 100, 200 |
| Quadruple | 3e-5 | 24 | 10 |
| COQE | 3e-5 | 32 | 100, 200 |

Table 6: Detailed Hyper-parameters.

the input format of RexUIE. We implement the pre-training model and trainer based on Transformers (Wolf et al., 2020). We adopt DeBERTaV3-Large (He et al., 2021) as our text encoder. We set the maximum token length to 512, and the maximum length of prompt to 256 . We split a query into sub-queries containing prompt text segments when the length of the prompt text is beyond the limit. Our model is optimized by AdamW (Loshchilov and Hutter, 2017), with weight decay as 0.01, threshold $\delta$ as 0. We set the clip gradient norm as 2, warmup ratio as 0.1. The hyper-parameters for grid search are listed in Table 6.

## C Details of Experiments Settings

**Few-Shot IE** We conducted few-shot experiments on one dataset for each task, following Lu et al. (2022) and Lou et al. (2023). Specifically, we sample 1/5/10 sentences for each type of entity/relation/event/sentiment from the training set. To avoid the influence of sampling noise, we repeated each experiment 10 times with different samples.

**Zero-Shot IE** We used CoNLL04 for RE due to its unique subject and object entity types for each relation. For NER, we employed CoNLLpp (Wang et al., 2019), which is a corrected version of the CoNLL2003 NER dataset. In order to prevent the performance of ChatGPT from being affected by randomly selected instructions, we adopted the SoTA zero-shot information extraction framework with ChatGPT proposed by Wei et al. (2023).

**Complex Schema Extraction** T5-UIE was initially limited to extracting only triples. We designed a forced approach to extract quintuples for T5-UIE, which extracts three tuples to form one quintuple.

The quintuple in COQE can be represented as (*subject, object, aspect, opinion, sentiment*). We

propose to model the quintuple extraction as extracting three triples: (*subject*, "subject-object", *object*), (*object*, "object-aspect", *aspect*), and (*aspect, sentiment, opinion*).

## D Ablation Study

|  | Method | Precision | Recall | F1 |
|---|---|---|---|---|
| Entity CoNLL03 | RexUIE | 92.95 | 93.66 | 93.31 |
|  | w/o PI | 93.00 | 93.79 | 93.39 |
|  | w/o Rt | 92.16 | 93.84 | 92.99 |
|  | w/o PI+Rt | 92.37 | 93.52 | 92.94 |
| Relation CoNLL04 | RexUIE | 78.80 | 74.88 | 76.79 |
|  | w/o PI | 75.31 | 72.27 | 73.76 |
|  | w/o Rt | 75.06 | 72.75 | 73.89 |
|  | w/o PI+Rt | 73.56 | 72.51 | 73.03 |
| Event Trigger ACE05-Evt | RexUIE | 71.36 | 75.24 | 73.25 |
|  | w/o PI | 73.80 | 72.41 | 73.10 |
|  | w/o Rt | 72.06 | 76.65 | 74.29 |
|  | w/o PI+Rt | 73.51 | 72.64 | 73.07 |
| Event Argument ACE05-Evt | RexUIE | 56.69 | 57.86 | 57.27 |
|  | w/o PI | 57.74 | 56.97 | 57.36 |
|  | w/o Rt | 57.93 | 59.64 | 58.77 |
|  | w/o PI+Rt | 55.84 | 55.34 | 55.59 |
| Sentiment 16-res | RexUIE | 75.18 | 81.32 | 78.13 |
|  | w/o PI | 78.45 | 78.60 | 78.52 |
|  | w/o Rt | 69.27 | 81.13 | 74.73 |
|  | w/o PI+Rt | 70.88 | 78.60 | 74.54 |
| AVG | RexUIE |  |  | **75.75** |
|  | w/o PI |  |  | 75.23 |
|  | w/o Rt |  |  | 74.93 |
|  | w/o PI+Rt |  |  | 73.83 |

Table 7: Ablation Study. PI denotes Prompts Isolation, Rt denotes rotary rmbedding. AVG is calculated over the five tasks.

We conducted an ablation experiment on RexUIE to explore the influence of Prompts Isolation and rotary embedding on the model, where RexUIE is not pre-trained. The results are listed in Table 7.

The experimental results demonstrate that removing Prompts Isolation leads to a decrease in the performance of RE, while the exclusion of rotary embedding results in detrimental effects on both relation and sentiment extraction. Overall, the complete RexUIE exhibits superior performance. Removing Prompts Isolation or rotary embedding results in a slight decline in performance, with the most significant drop observed when both are deleted.

## E Influence of Encoder

DeBERTa (He et al., 2021) improved BERT (Devlin et al., 2019) and RoBERTa (Liu et al., 2019)

| Dataset | USM | RexUIE$_{RoBERTa}$ | RexUIE |
|---|---|---|---|
| CoNLL03 | 92.76 | 92.98 | **93.31** |
| CoNLL04 | 75.86 | **77.15** | 76.79 |
| ACE05-Evt (Trigger) | 71.68 | 72.15 | **73.25** |
| ACE05-Evt (Argument) | 55.37 | **57.77** | 57.27 |
| 16-res | 76.73 | 77.13 | **78.13** |
| Average | 74.48 | 75.44 | **75.75** |

Table 8: Replacing DeBERTa with RoBERTa. We compare RexUIE$_{RoBERTa}$ with USM as they share the same encoder structure.

models using the disentangled attention mechanism and enhanced mask decoder. Corrected sentence: To ensure that our work was not entirely dependent on a better text encoder, we replaced DeBERTaV3-Large with RoBERTa-Large, denoted as RexUIE$_{RoBERTa}$, thus maintaining the same text encoder settings as USM. We present the comparison in Table 8.

RexUIE$_{RoBERTa}$ shows a performance level between USM and RexUIE, demonstrating that our proposed approach does indeed improve performance, and incorporating DeBERTaV3 to RexUIE further enhances this improvement.

## F   Insights to the Schema Complexity and Training Data Size

Under the full-shot setting, the improvement of RexUIE compared to the previous UIE models is not significant (only 1% across 4 tasks and 14 metrics). At the same time, we have also found that for different tasks or datasets, the improvement of RexUIE seems to exhibit some randomness, which may be related to several factors such as schema complexity, training data size, task type, and the extent of task exploration. Among these factors, we believe that schema complexity and training data size are more important, so we conducted a statistical analysis to better summarize the patterns. (We only consider the case of full-shot without pre-training to avoid the influence of pre-training.)

- Schema complexity: Due to ESI and recursive strategies, we intuitively believe that RexUIE has certain advantages in handling complex schemas. We use the number of leaf nodes in the schema to represent the complexity of the schema, noted as $C$.

- Training data size: We know that as the training data size increases, the differences between the performance of models will be nar-

row. Therefore, we believe that the performance improvement is negatively correlated with training data size. We note the training data size as $S$.

To investigate the pattern, we introduce a media variable $\log(10000 \times \frac{C}{S})$. After removing certain outliers and event-trigger datasets, we find a positive correlation between $\log(10000 \times \frac{C}{S})$ and the relative improvement in Table 9, which supports our hypothesis.

## G   Pre-training Data

$\mathcal{D}_{distant}$   We remove abstract and over-specialized entity types and relation types (such as "structural class of chemical compound") and remove categories that occur less than 10000 times. We also remove the examples that do not contain any relations. Finally, we collect 3M samples containing entities and relations as $\mathcal{D}_{distant}$.

$\mathcal{D}_{superv}$   We collect some high-quality data for named entity recognition and relationship extraction from publicly available open domain supervised datasets. Specifically, we employ OntoNotes (Pradhan et al., 2013), NYT (Riedel et al., 2013), CrossNER (Liu et al., 2020), Few-NERD (Ding et al., 2021), kbp37 (Zhang and Wang, 2015), Mit Restaurant and Movie corpus (Liu et al., 2013) together as $\mathcal{D}_{superv}$.

$\mathcal{D}_{mrc}$   Specifically, we collect SQuAD (Rajpurkar et al., 2016) and HellaSwag (Zellers et al., 2019) together as $\mathcal{D}_{mrc}$. The MRC data is constructed with pairs of questions and answers. For implementations, we use the question as the type, and consider the answer as the span to extract.

## H   Example of Schema

Schema examples for some datasets are listed in Table 10.

## I   Query Example

Some query examples are listed in Table 11 and Table 12.

| Dataset | Schema Complexity $C$ | Training Data Size $S$ | $\log(10000 \times \frac{C}{S})$ | USM | RexUIE | Relative Improvement |
|---|---|---|---|---|---|---|
| CoNLL03 | 7 | 14041 | 0.6977 | 92.79 | 93.31 | 0.56% |
| ACE04 | 4 | 6202 | 0.8095 | 87.79 | 88.02 | 0.26% |
| ACE05-Ent | 7 | 7299 | 0.9818 | 86.98 | 86.87 | -0.13% |
| NYT | 55 | 56196 | 0.9907 | 94.07 | 94.55 | 0.51% |
| 14-res | 3 | 1266 | 1.3747 | 76.35 | 76.36 | 0.01% |
| 14-lap | 3 | 906 | 1.5200 | 65.46 | 66.92 | 2.23% |
| 16-res | 3 | 857 | 1.5441 | 76.73 | 78.13 | 1.82% |
| ACE05-Evt-Arg | 79 | 19216 | 1.6140 | 55.37 | 57.27 | 3.43% |
| 15-res | 3 | 605 | 1.6954 | 68.8 | 70.48 | 2.44% |
| CoNLL04 | 6 | 922 | 1.8134 | 75.86 | 76.79 | 1.23% |
| SciERC | 115 | 1861 | 2.7910 | 37.05 | 38.16 | 3.00% |

Table 9: The correlation between schema complexity, training data size and relative improvement.

| Dataset | Schema **C** | Example of **t** |
|---|---|---|
| Entity CoNLL03 | {"person": null, "location": null, "miscellaneous": null, "organization": null} | ["person"] |
| Relation CoNLL04 | {"organization": {"organization in ( location )": null}, "other": null, "location": {"located in ( location )": null}, "people": {"live in ( location )": null, "work for ( organization )": null, "kill ( people )": null}} | ["organization", "organization in ( location )"] |
| Event ACE05-Evt | {"attack": {"attacker": null, "place": null, "target": null, "instrument": null}, "end position": {"person": null, "place": null, "entity": null}, "meet": {"place": null, "entity": null}, "transport": {"artifact": null, "origin": null, "destination": null, "agent": null, "vehicle": null}, "die": {"victim": null, "instrument": null, "place": null, "agent": null}, "transfer money": {"giver": null, "beneficiary": null, "recipient": null}, "trial hearing": {"adjudicator": null, "defendant": null, "place": null}, "charge indict": {"defendant": null, "place": null, "adjudicator": null}, "transfer ownership": {"beneficiary": null, "artifact": null, "seller": null, "place": null, "buyer": null}, "sentence": {"defendant": null, "place": null, "adjudicator": null}, "extradite": {"person": null, "destination": null, "agent": null}, "start position": {"place": null, "entity": null, "person": null}, "start organization": {"organization": null, "agent": null, "place": null}, "sue": {"defendant": null, "plaintiff": null}, "divorce": {"person": null, "place": null}, "marry": {"person": null, "place": null}, "phone write": {"place": null, "entity": null}, "injure": {"victim": null}, "end organization": {"organization": null}, "appeal": {"adjudicator": null, "plaintiff": null}, "convict": {"defendant": null, "place": null, "adjudicator": null}, "fine": {"entity": null, "adjudicator": null}, "declare bankruptcy": {"organization": null}, "demonstrate": {"place": null, "entity": null}, "elect": {"person": null, "place": null, "entity": null}, "nominate": {"person": null}, "acquit": {"defendant": null, "adjudicator": null}, "execute": {"agent": null, "person": null, "place": null}, "release parole": {"person": null}, "arrest jail": {"person": null, "place": null, "agent": null}, "born": {"person": null, "place": null}} | ["attack", "attacker"] |
| Sentiment 16-res | {"aspect": {"positive ( opinion )": null, "neutral ( opinion )": null, "negative ( opinion )": null}, "opinion": null} | ["aspect", "positive ( opinion )"] |
| COQE Camera | {"subject": {"object": {"aspect": {"worse ( opionion )": null, "equal ( opinion )": null, "better ( opinion )": null, "different ( opinion )": null}, "worse ( opionion )": null, "equal ( opinion )": null, "better ( opinion )": null, "different ( opinion )": null}, "aspect": {"worse ( opionion )": null, "equal ( opinion )": null, "better ( opinion )": null, "different ( opinion )": null}, "worse ( opionion )": null, "equal ( opinion )": null, "better ( opinion )": null, "different ( opinion )": null}, "object": {"aspect": {"worse ( opionion )": null, "equal ( opinion )": null, "better ( opinion )": null, "different ( opinion )": null}, "worse ( opionion )": null, "equal ( opinion )": null, "better ( opinion )": null, "different ( opinion )": null}, "aspect": {"worse ( opionion )": null, "equal ( opinion )": null, "better ( opinion )": null, "different ( opinion )": null}, "worse ( opionion )": null, "equal ( opinion )": null, "better ( opinion )": null, "different ( opinion )": null}} | ["subject", "object", "aspect", "better ( opinion )"] |

Table 10: Schema examples.

| Dataset | Sample Id | Query |
|---------|-----------|-------|
| CoNLL03 | 0 | [CLS][P][T] location[T] miscellaneous[T] organization[T] person[Text] EU rejects German call to boycott British lamb .[SEP] |
| CoNLL04 | 0 | [CLS][P][T] location[T] organization[T] other[T] people[Text] The self-propelled rig Avco 5 was headed to shore with 14 people aboard early Monday when it capsized about 20 miles off the Louisiana coast , near Morgan City , Lifa said.[SEP] |
| | | [CLS][P] location: Morgan City[T] located in ( location )[P] location: Louisiana[T] located in ( location )[P] people: Lifa[T] kill ( people )[T] live in ( location )[T] work for ( organization )[Text] The self-propelled rig Avco 5 was headed to shore with 14 people aboard early Monday when it capsized about 20 miles off the Louisiana coast , near Morgan City , Lifa said.[SEP] |
| ACE05-Evt | 0 | [CLS][P][T] acquit[T] appeal[T] arrest jail[T] attack[T] born[T] charge indict[T] convict[T] declare bankruptcy[T] demonstrate[T] die[T] divorce[T] elect[T] end organization[T] end position[T] execute[T] extradite[T] fine[T] injure[T] marry[T] meet[T] merge organization[T] nominate[T] pardon[T] phone write[T] release parole[T] sentence[T] start organization[T] start position[T] sue[T] transfer money[T] transfer ownership[T] transport[T] trial hearing[Text] The electricity that Enron produced was so exorbitant that the government decided it was cheaper not to buy electricity and pay Enron the mandatory fixed charges specified in the contract .[SEP] |
| | | [CLS][P] transfer money: pay[T] beneficiary[T] giver[T] place[T] recipient[Text] The electricity that Enron produced was so exorbitant that the government decided it was cheaper not to buy electricity and pay Enron the mandatory fixed charges specified in the contract .[SEP] |
| | 1 | [CLS][P][T] acquit[T] appeal[T] arrest jail[T] attack[T] born[T] charge indict[T] convict[T] declare bankruptcy[T] demonstrate[T] die[T] divorce[T] elect[T] end organization[T] end position[T] execute[T] extradite[T] fine[T] injure[T] marry[T] meet[T] merge organization[T] nominate[T] pardon[T] phone write[T] release parole[T] sentence[T] start organization[T] start position[T] sue[T] transfer money[T] transfer ownership[T] transport[T] trial hearing[Text] and he has made the point repeatedly in interview after interview that he has never claimed to speak for god , nor has he claimed that this is " god ś war "[SEP] |
| | | [CLS][P] attack: war[T] attacker[T] instrument[T] place[T] target[T] victim[Text] and he has made the point repeatedly in interview after interview that he has never claimed to speak for god , nor has he claimed that this is " god ś war "[SEP] |

Table 11: Query examples for CoNLL03, CoNLL04 and ACE05-Evt.

| Dataset | Sample Id | Query |
|---|---|---|
| 16-res | 0 | [CLS][P][T] aspect[T] opinion[Text] Judging from previous posts this used to be a good place , but not any longer .[SEP] |
| | | [CLS][P] aspect: place[T] negative ( opinion )[T] neutral ( opinion )[T] positive ( opinion )[Text] Judging from previous posts this used to be a good place , but not any longer .[SEP] |
| | 1 | [CLS][P][T] aspect[T] opinion[Text] The food was lousy - too sweet or too salty and the portions tiny .[SEP] |
| | | [CLS][P] aspect: portions[T] negative ( opinion )[T] neutral ( opinion )[T] positive ( opinion )[P] aspect: food[T] negative ( opinion )[T] neutral ( opinion )[T] positive ( opinion )[Text] The food was lousy - too sweet or too salty and the portions tiny .[SEP] |
| COQE-Camera | 0 | [CLS][P][T] aspect[T] better ( opinion )[T] different ( opinion )[T] equal (opinion )[T] object[T] subject[T] worse ( opionion )[Text] Also , both the Nikon D50 and D70S will provide sharper pictures with better color saturation and contrast right out of the camera .[SEP] |
| | | [CLS][P] subject: Nikon D50[T] aspect[T] better ( opinion )[T] different ( opinion )[T] equal (opinion )[T] object[T] worse ( opionion )[P] subject: D70S[T] aspect[T] better ( opinion )[T] different ( opinion )[T] equal (opinion )[T] object[T] worse ( opionion )[Text] Also , both the Nikon D50 and D70S will provide sharper pictures with better color saturation and contrast right out of the camera .[SEP] |
| | | [CLS][P] subject: D70S,aspect: pictures[T] better ( opinion )[T] different ( opinion )[T] equal (opinion )[T] worse ( opionion )[P] subject: D70S,aspect: color saturation[T] better ( opinion )[T] different ( opinion )[T] equal (opinion )[T] worse ( opionion )[P] subject: Nikon D50,aspect: pictures[T] better ( opinion )[T] different ( opinion )[T] equal (opinion )[T] worse ( opionion )[P] subject: Nikon D50,aspect: contrast[T] better ( opinion )[T] different ( opinion )[T] equal (opinion )[T] worse ( opionion )[P] subject: Nikon D50,aspect: color saturation[T] better ( opinion )[T] different ( opinion )[T] equal (opinion )[T] worse ( opionion )[P] subject: D70S,aspect: contrast[T] better ( opinion )[T] different ( opinion )[T] equal (opinion )[T] worse ( opionion )[Text] Also , both the Nikon D50 and D70S will provide sharper pictures with better color saturation and contrast right out of the camera .[SEP] |

Table 12: Query examples for 16-res and COQE-Camera.