# OpenReview forum: "RexUIE: A Recursive Method with Explicit Schema Instructor for Universal Information Extraction"
_EMNLP/2023/Conference — EMNLP 2023 Findings_

### Official Review · Reviewer_SsUQ · 2023-07-21

**Typos Grammar Style And Presentation Improvements:** 1. I personally think that the discon…
**Soundness:** 4

**Excitement:**

3: Ambivalent: It has merits (e.g., it reports state-of-the-art results, the idea is nice), but there are key weaknesses (e.g., it describes incremental work), and it can significantly benefit from another round of revision. However, I won't object to accepting it if my co-reviewers champion it.

**Missing References:**

CasRel[1] proposed using consistent tagging schemes to model the extraction of entities and relationships, and TPlinker[2] extended the idea to a unified matrix representation. CasRel was not cited in this paper, and while TPlinker was mentioned at line 269, no corresponding description was provided. Although they did not mention the extraction of multi-granular structures like events, the idea of transforming multi-granular structures into combinations of binary relations has been discussed in NLP research for a long time. Therefore, I suggest introducing CasRel and TPlinker as related works in joint extraction in Section 2.

[1] Zhepei Wei, Jianlin Su, Yue Wang, Yuan Tian, and Yi Chang. 2020. A Novel Cascade Binary Tagging Framework for Relational Triple Extraction. In Proceedings of the 58th Annual Meeting of the Association for Computational Linguistics, pages 1476–1488, Online. Association for Computational Linguistics.

[2] Yucheng Wang, Bowen Yu, Yueyang Zhang, Tingwen Liu, Hongsong Zhu, and Limin Sun. 2020. Tplinker: Single-stage joint extraction of entities and relations through token pair linking.

**Paper Topic And Main Contributions:**

This paper proposes an improved version of UIE, the Unified Structure Generation approach. It replaces the structural schema instructor in UIE with an explicit schema instructor where not only the schema terms but also their structures are provided. This paper proposes the idea of recursively extracting aspects instead of doing so separately. The results are significantly improved on most of the evaluated benchmarks.

**Questions For The Authors:**

Question A: Does the premise for judging how to perform recursive extraction, the Explicit Schema Instructor, implies prior knowledge of all possible schemas?

Question B: Is the explicit schema information also provide to compared methods?  Could the author provide a more clearly explanation about how much improvement is solely due to a more complete schema, and how much is attributed to the advancement in recursive extraction techniques?

Question C: Are there any instance of incomplete schemas in practice? For example, the schema requires the extraction result to have four aspects, but only three of them appear in the utterance. Will the proposed method work correctly in such cases?

Question D: CasRel and TPlinker (please refer to missing references) proposed a matrix-based tagging scheme partly to handle the phenomenon of overlapping relations in the extraction task. (For instance, in the sentence "A and B boarded the plane together," two facts can be extracted: "A boarded the plane" and "B boarded the plane," resulting in the plane being used twice). This work strictly follows the structure of an explicit schema for extraction. Does it also have the capability to handle overlapping relations?

**Reasons To Accept:**

1. The improvements are significaint.
2. The writing is fluent, making the paper easy to read.
3. The idea of recursive extraction is novel in joint extraction methods with matrix-based tagging schemes.

**Reasons To Reject:**

1. The reasons behind the results improvement might require further discussion. (Please refer to Question B)
2. The requirement for explicit schema may limit the practicality of this method. (Please refer to Question C and D)

**Reproducibility:**

5: Could easily reproduce the results.

**Reviewer Confidence:**

5: Positive that my evaluation is correct. I read the paper very carefully and I am very familiar with related work.

---

> ### Author Rebuttal · Authors · 2023-08-27
>
> **Thank you for your comments. And we appreciate you for referring to the questions in your concerns. This allows us to address them more effectively.**
>
> Before answering your questions, we would like to clarify the difference between Explicit Schema Instructor (ESI) and Implicit Schema Instructor (ISI) more clearly. Taking relation extraction as an example, in previous UIE works (T5-UIE, USM), the Implicit Schema Instructor simply concatenats all entity types and relation types without explicitly representing the constraints between entity types and relation types. This means that the model can only implicitly learn the constraints between entity types and relation types through training. However, our method, RexUIE, **explicitly prevents the model from generating illegal matches between entity types and relation types through the behavior-level constraints** introduced by the recursive extraction strategy. This is the fundamental difference between ESI and ISI.
>
> As you mentioned in Question A, ESI determines how to perform recursive extraction. Therefore, **ESI and the recursive inference strategy are tightly integrated**. We cannot separate ESI from the recursive extraction strategy to evaluate its contribution to the performance of RexUIE.
>
> ## Question A: Does the premise for judging how to perform recursive extraction, the Explicit Schema Instructor, implies prior knowledge of all possible schemas?
>
> Yes. Currently, the academic definition of the IE task is to extract information tuples that satisfy the schema constraints from a given text. We believe that the utilization of ESI and the requirement of prior knowledge of all possible schemas are unrelated. The definition of the IE tasks determines that we need to know all possible schemas in advance, and both T5-UIE and USM (using ISI) require prior knowledge of all possible schemas as well.
>
> ## Question B: Is the explicit schema information also provide to compared methods? Could the author provide a more clearly explanation about how much improvement is solely due to a more complete schema, and how much is attributed to the advancement in recursive extraction techniques?
>
> This is a very insightful question. We also considered this question when conducting ablation experiments. As you have mentioned in Question A, ESI determines how recursive extraction is performed, indicating that **ESI and recursive reasoning are closely intertwined**. However, since T5-UIE and USM do not utilize recursive reasoning, we were unable to incorporate ESI into their models. Additionally, the inseparability between ESI and recursion makes it difficult to quantify the individual contribution of each to RexUIE.
>
> Nonetheless, in order to demonstrate the effectiveness of ESI as much as possible, we analyzed the distribution of "relation types" versus "subject type-object type" predicted by T5-UIE, which uses Implicit Schema Instructor, in Section 5.6. We discovered that T5-UIE often generates outputs that do not adhere to the schema constraints in low-resource settings.
>
> ## Question C: Are there any instance of incomplete schemas in practice? For example, the schema requires the extraction result to have four aspects, but only three of them appear in the utterance. Will the proposed method work correctly in such cases?
>
> This is a very insightful question as well. First of all, please allow us to reiterate the definition of IE. Currently, the academic definition of the IE is to extract information tuples from a given text that satisfy the schema constraints. Therefore, it is invalid if a model extracts tuples of information out of the schema.
>
> Regrading to your question, **RexUIE can extract instances of incomplete schemas**. In fact, the case you mentioned exists in COQE. Sometimes, not all elements of (subject, object, aspect, opinion, polarity) present in the sentence, and the task allows for the omission of arbitrary spans. When conducting experiments on this dataset, we found that RexUIE can effectively handle these cases. There is a showcase below, where the object (another camera) is omitted in the context:
> > By the way you can use the same battery the S50 comes with .
> >
> > subject: "S50"; aspect: "battery"; equal ( opinion ): "same"
>
> In Appendix G (Table 9), we demonstrate how to write the schema to solve the problem. The results of COQE in Table 1 also demonstrate that RexUIE achieves good performance in the cases you mentioned.
>
> ## Question D: CasRel and TPlinker (please refer to missing references) proposed a matrix-based tagging scheme partly to handle the phenomenon of overlapping relations in the extraction task. (For instance, in the sentence "A and B boarded the plane together," two facts can be extracted: "A boarded the plane" and "B boarded the plane," resulting in the plane being used twice). This work strictly follows the structure of an explicit schema for extraction. Does it also have the capability to handle overlapping relations?
>
> Yes, RexUIE has the ability to handle overlapping spans, including SEO and EPO [1]. In the case you mentioned:
> 1. In the first step, we input the Person type and extract (Person: A) and (Person: B).
> 2. In the second step, we input (Person: A, board) and (Person: B, board) and extract (Person: A, board: plane) and (Person: B, board: plane).
>
> [1] Extracting Relational Facts by an End-to-End Neural Model with Copy Mechanism.
>
> ## Overall
>
> Based on the answers to Question A, C and D, we firmly believe that compared with previous works, the practicality of RexUIE not only will not be limited by **explicit schema instructor**, but also enhanced due to the ability to support complex schemas, uncomplete schemas, et al.
>
> ## Missing References
> We will add the missing references to the final version. Thanks for reminding us.
>
> ## Typos Grammar Style And Presentation Improvements
> We will correct the typos and grammar style, and presentation issues in the final version. Thanks again!

---

### Official Review · Reviewer_zhWZ · 2023-08-02

**Soundness:** 4

**Excitement:**

4: Strong: This paper deepens the understanding of some phenomenon or lowers the barriers to an existing research direction.

**Paper Topic And Main Contributions:**

This paper proposes RexUIE, a novel approach for universal information extraction (UIE) that can extract complex schemas beyond just entity pairs. The key contributions are:

- Redefining UIE to cover extraction of any schema with multiple spans/types
- Introducing a recursive method to query all schema types and compute results via unified token-linking operations
- Using explicit schema instructions (ESI) to provide richer label semantics and enhance low-resource performance.
- Demonstrating state-of-the-art performance on simultaneous extraction of entities, relations, events, sentiment analysis, and complex schemas like quadruples and quintuples

**Questions For The Authors:**

Question A: l.72: "almost all extraction schemas": What are the schemas not covered?

Question B: What schemas or examples still challenge RexUIE?

Question C: You mentioned the gains over prior UIE models were relatively small in some cases. Do you have insights into why the improvements were marginal for certain tasks/datasets?

Question D: Have you explored using external knowledge bases or ontologies during pre-training to further enhance the model's semantic understanding? This could help boost low-resource performance.

Question E: The zero-shot evaluation was limited to NER and RE. What challenges do you foresee in extending zero-shot capabilities to events, sentiments or other tasks?

Question F: Has any analysis been done on how the size and richness of the schema affects RexUIE's performance? E.g. very large or nested schemas.

**Reasons To Accept:**

- The recursive method and explicit schema instructions are intuitive extensions for improving universal extraction. I liked the idea of using a schema when possible.
- Extensive experiments validate RexUIE's effectiveness on diverse datasets, especially for complex schemas.
- RexUIE improves over prior UIE models like T5-UIE and USM in both full-shot and few-shot settings
- The pre-training strategy is sensible for further boosting low-resource performance
- Well-written

**Reasons To Reject:**

- The approach is complex with many components (encoder, ESI, token operations, pre-training etc.) making ablation analysis difficult
- The gains over prior UIE models, while consistent, are relatively small in some cases
- Zero-shot capabilities only demonstrated for NER and RE but not events or sentiment

**Reproducibility:**

5: Could easily reproduce the results.

**Reviewer Confidence:**

3: Pretty sure, but there's a chance I missed something. Although I have a good feel for this area in general, I did not carefully check the paper's details, e.g., the math, experimental design, or novelty.

**Typos Grammar Style And Presentation Improvements:**

- When citing work, add the missing space before opening parenthesis (everywhere in the paper, eg, lines 138 or 959)
- l.474: according to [Wikipedia](https://en.wikipedia.org/wiki/Soviet_(council)), "Soviet" could be considered as an organisation

---

> ### Author Rebuttal · Authors · 2023-08-27
>
> We sincerely appreciate your diligent efforts and valuable suggestions. We will address your questions and concerns in a thorough manner.
>
> # Concerns
>
> ## The approach is complex with many components (encoder, ESI, token operations, pre-training etc.) making ablation analysis difficult
>
> This is a particularly good question, and when conducting the RexUIE work, we also devoted significant thought to devising effective ablation experiments.
>
> 1. Encoder: In order to remove the influence of schema order, we have adopted the Prompt Isolation strategy, and we believe that relative positional encoding would be more advantageous for prompt isolation. Therefore, we have utilized DeBERTa as our encoder. **To eliminate the impact of encoder variation, we have conducted a performance comparison with the same base model, which is presented in Table 8 (Appendix E).** Our model based on RoBERTa-Large outperforms USM by nearly 1 point and only lags behind RexUIE based on DeBERTa-Large by 0.31 points in the full-shot scenario. Thus, we contend that the architecture has played a more significant role in enhancing performance.
> 2. ESI & Token Operation: As you mentioned, the Explicit Schema Instructor is an intuitive extension aimed at enhancing UIE performance under low-resource settings. In Section 5.6, we examined the influence of this module by analyzing the distribution of "relation type" versus "subject type-object type" predicted by T5-UIE, which utilizes the Implicit Schema Instructor. We observed that under low-resource setting, T5-UIE often generates outputs that do not adhere to the schema constraints. However, as you correctly pointed out, quantifying the impact of the Explicit Schema Instructor and our recursive strategy, including Token Operation, **presents a significant challenge due to their inseparability**.
> 3. Pre-training: Since that T5-UIE and USM did not release their pre-training datasets, we can only strive to keep our pre-training data as consistent as possible with theirs.
>
> ## The gains over prior UIE models, while consistent, are relatively small in some cases
>
> Under the full-shot setting, the improvement of RexUIE compared to previous UIE models is not particularly significant (1% across 4 tasks and 14 metrics). We consider that the main reason is that the datasets we used in the experiment are **suffiently explored**.
>
> Additionally, by checking the experimental results in Table 1 and Table 2, we report the average improvement of RexUIE over USM in the following Table, from which we can observe that **as the training data increases, the relative improvement gradually decreases**.
>
> |Datasets of Table 2|1-shot|5-shot|10-shot|Full-shot|
> |-|-|-|-|-|
> |AVG Improvement RexUIE/USM|32.9|6.7|6.0|1.3|
>
> A more detailed analysis is provided in the answer to Question C.
>
> ## Zero-shot capabilities only demonstrated for NER and RE but not events or sentiment
>
> This is also an insightful question. There are two reasons why we did not report the zero-shot performance of RexUIE for EE or ABSA:
> 1. We attempted to compare our work with previous studies in terms of EE and ABSA datasets. However, since **USM did not report these experiments and they did not provide code**, it was challenging for us to find a suitable UIE baseline.
> 2. As mentioned in Section "Limitations", we intuitively believe that the performance of RexUIE in EE and ABSA would not be as ideal as NER and RE due to **the limitation of pre-training data**.
>
> We will explore the zero-shot capabilities for more IE tasks if we find a way to solve the problems above.
>
> # Questions
> ## Question A: l.72: "almost all extraction schemas": What are the schemas not covered?
>
> Our current approach is not yet capable of solving discontiguous entity extraction and open information extraction.
>
> ## Question B: What schemas or examples still challenge RexUIE?
>
> - Challenging Schemas: discontiguous entity extraction and open information extraction.
> - Challenging Examples: examples with very long context or challenging schemas.
>
> ## Question C: You mentioned the gains over prior UIE models were relatively small in some cases. Do you have insights into why the improvements were marginal for certain tasks/datasets?
>
> Very insightful question. Under the full-shot setting, the improvement of RexUIE compared to the previous UIE models is not significant (only 1% across 4 tasks and 14 metrics). At the same time, we have also found that for different tasks or datasets, the improvement of RexUIE seems to exhibit some randomness, which may be related to several factors such as schema complexity, training data size, task type, and the extent of task exploration. Among these factors, we believe that **schema complexity and training data size** are more important, so we conducted a statistical analysis to better summarize the patterns. (We only consider the case of full-shot without pre-training to avoid the influence of pre-training.)
> 1. Schema complexity: Due to ESI and recursive strategies, we intuitively believe that RexUIE has certain advantages in handling complex schemas. We use the number of leaf nodes in the schema to represent the complexity of the schema, noted as $C$.
> 2. Training data size: We know that as the training data size increases, the differences between the performance of models will be narrow. Therefore, we believe that the performance improvement is negatively correlated with training data size. We note the training data size as $S$.
>
> To illustrate this pattern, we have created a line plot that visually demonstrates the relationship between the improvement $R$ and $\log(10000\times C / S)$ (unfortunately, images are not allowed to appear during rebuttal). After removing certain outliers and event-trigger datasets, we find a clear **positive correlation between $R$ and $\log(10000\times C / S)$**，which supports our hypothesis. The results are consistent with our analysis and will be included in the final version.
>
> The values of $R$ and $\log(10000\times C / S)$ across various datasets are shown in the Table below.
>
> ||Schema Complexity $C$|Training Data Size $S$|$\log(10000\times C / S)$|USM|RexUIE|Relative Improvement $R$|
> |-|-|-|-|-|-|-|
> |CoNLL03|7|14041|0.6977 |92.79|93.31|0.56%|
> |ACE04|4|6202|0.8095|87.79|88.02|0.26%|
> |ACE05-Ent|7|7299|0.9818|86.98|86.87|-0.13%|
> |NYT|55|56196|0.9907|94.07|94.55|0.51%|
> |14-res|3|1266|1.3747|76.35|76.36|0.01%|
> |14-lap|3|906|1.5200|65.46|66.92|2.23%|
> |16-res|3|857|1.5441|76.73|78.13|1.82%|
> |ACE05-Evt-Arg|79|19216|1.6140|55.37|57.27|3.43%|
> |15-res|3|605|1.6954|68.8|70.48|2.44%|
> |CoNLL04|6|922|1.8134|75.86|76.79|1.23%|
> |SciERC|115|1861|2.7910|37.05|38.16|3.00%|
>
> ## Question D: Have you explored using external knowledge bases or ontologies during pre-training to further enhance the model's semantic understanding? This could help boost low-resource performance.
>
> We have not explored using external knowledge bases or ontologies, but it is really a promising suggestion. We will explore it in the future work.
>
> ## Question E: The zero-shot evaluation was limited to NER and RE. What challenges do you foresee in extending zero-shot capabilities to events, sentiments or other tasks?
>
> As our response to concern 3, we have not extended zero-shot capabilities to events and sentiments due to the lack of baselines and pre-training data.
>
> ## Question F: Has any analysis been done on how the size and richness of the schema affects RexUIE's performance? E.g. very large or nested schemas.
>
> We conducted a statistical analysis of the relative and absolute performance changes of RexUIE with the increasing complexity of schema (Unfortunately, images are not allowed to appear during rebuttal. It will be added to the final version). The following observations can be made:
> 1. **As the Schema Complexity increases, the relative performance of RexUIE compared to USM improves**, which also demonstrates the advantage of RexUIE over previous work in solving complex schemas.
> 2. **When the Schema Complexity raises up to a certain level, the absolute performance of RexUIE decreases**, which we consider to be a reasonable result since higher complexity increases the difficulty of the task.
>
> We appreciate your careful review and notice. We will correct the typos and grammar style, and presentation issues in the final version.
>
> **Overall, we would like to express our gratitude for your recognition of our work and for providing such a detailed evaluation and insightful questions. We hope that our responses can alleviate any doubts you may have.**

---

### Official Review · Reviewer_zUQ9 · 2023-08-06

**Soundness:** 4

**Excitement:**

3: Ambivalent: It has merits (e.g., it reports state-of-the-art results, the idea is nice), but there are key weaknesses (e.g., it describes incremental work), and it can significantly benefit from another round of revision. However, I won't object to accepting it if my co-reviewers champion it.

**Paper Topic And Main Contributions:**

This paper proposes a new model for Universal Information Extraction (UIE). Compared to previous UIE models, this model includes a novel component called "Explicit Schema Instructor" (ESI), which is the concatenation of extraction results in the previous iteration and the extraction schema. This component allows the model to perform IE under almost all extraction schema, and the model shows great performance on 17 different datasets across 6 different tasks.

**Reasons To Accept:**

1. The paper proposes a new model for Universal Information Extraction (UIE), which shows strong empirical performance.
2. The authors conduct extensive experiments in various settings to demonstrate the effectiveness of the proposed model.

**Reasons To Reject:**

1. The performance improvement over baselines is hard to interpret since the proposed model uses a different base model (DeBERTa) apart from the architecture change.
2. The proposed method includes a flexible architecture to unify a couple of IE tasks but it cannot do real zero-shot extraction with unseen schemas.

**Reproducibility:**

4: Could mostly reproduce the results, but there may be some variation because of sample variance or minor variations in their interpretation of the protocol or method.

**Reviewer Confidence:**

4: Quite sure. I tried to check the important points carefully. It's unlikely, though conceivable, that I missed something that should affect my ratings.

---

> ### Author Rebuttal · Authors · 2023-08-27
>
> We appreciate your recognition of the novelty and experimental effectiveness of RexUIE. Indeed, UIE is a field that requires extensive experiments across various settings. Thus, we conducted as comprehensive experiments as possible within our scope of considerations. We are delighted that you recognize our work.
>
> We also appreciate your valuable suggestions, and we will provide explanations point by point to address your concerns. We hope that we can alleviate any worries you may have.
>
> > 1. The performance improvement over baselines is hard to interpret since the proposed model uses a different base model (DeBERTa) apart from the architecture change.
>
> Due to space limitations, **we have presented the performance comparison with the same base model in Table 8 (Appendix E)**. As shown in Table 8, based on RoBERTa-Large, our model outperforms USM by nearly 1 point, and falls slightly behind RexUIE based on DeBERTa-Large by 0.31 points under the full-shot setting. Therefore, **we believe that the architecture of RexUIE plays a more substantial role in enhancing performance**.
>
> We genuinely appreciate your valuable suggestions, and we will provide a clearer explanation of these comparative experiments in the main body of our paper.
>
> > 2. The proposed method includes a flexible architecture to unify a couple of IE tasks but it cannot do real zero-shot extraction with unseen schemas.
>
> **In Section 5.4, we conducted zero-shot experiments for the NER and RE tasks.** As illustrated in Table 3, RexUIE exhibited a significant improvement over both USM and ChatIE. To ensure the fairness of the experiments, we **maintained a consistent zero-shot setting with USM** and deliberately excluded the data from Table 3 during the pre-training process.

---

### Meta-Review · Area_Chair_xkjE · 2023-09-20

**Recommendation:** 4

**Metareview:**

The paper proposes a new model for Universal Information Extraction (UIE) named RexUIE, which concatenates extraction results from the previous iteration with the extraction schema. The proposed methods are evaluated across 17 different datasets and 6 different tasks, such as extracting entities, relations, events, sentiment, and quadruples and quintuples.

Overall, the proposed model demonstrates a minor performance gain over the baseline in some cases when the entire training dataset is used (full-shot training), but achieves higher performance gain under few-shot settings. The paper is well-written and easy to follow, and the proposed idea is novel. There are a few missing references, and a table does not follow the EMNLP template, but these issues will be fixed in the final version.

---

### Decision · Program_Chairs · 2023-10-07

**Decision:**

Accept-Findings

**Comment:**

The paper proposes a new model for Universal Information Extraction (UIE) named RexUIE, which concatenates extraction results from the previous iteration with the extraction schema. The proposed methods are evaluated across 17 different datasets and 6 different tasks, such as extracting entities, relations, events, sentiment, and quadruples and quintuples.

Overall, the proposed model demonstrates a minor performance gain over the baseline in some cases when the entire training dataset is used (full-shot training), but achieves higher performance gain under few-shot settings. The paper is well-written and easy to follow, and the proposed idea is novel. There are a few missing references, and a table does not follow the EMNLP template, but these issues will be fixed in the final version.